# Plasma Metabolite Response to Simple, Refined and Unrefined Carbohydrate-Enriched Diets in Older Adults—Randomized Controlled Crossover Trial

**DOI:** 10.3390/metabo12060547

**Published:** 2022-06-15

**Authors:** Neil K. Huang, Nirupa R. Matthan, Gregory Matuszek, Alice H. Lichtenstein

**Affiliations:** 1Cardiovascular Nutrition Laboratory, Jean Mayer USDA Human Nutrition Research Center on Aging, Tufts University, Boston, MA 02111, USA; neil.huang@tufts.edu (N.K.H.); nirupa.matthan@tufts.edu (N.R.M.); 2Informatics Core Unit, Jean Mayer USDA Human Nutrition Research Center on Aging, Tufts University, Boston, MA 02111, USA; gregory.matuszek@tufts.edu

**Keywords:** simple carbohydrate, refined carbohydrate, unrefined carbohydrate, diet, biomarkers, randomized controlled crossover feeding trial

## Abstract

Food intake data collected using subjective tools are prone to inaccuracies and biases. An objective assessment of food intake, such as metabolomic profiling, may offer a more accurate method if unique metabolites can be identified. To explore this option, we used samples generated from a randomized and controlled cross-over trial during which participants (*N* = 10; 65 ± 8 year, BMI, 29.8 ± 3.2 kg/m^2^) consumed each of the three diets enriched in different types of carbohydrate. Plasma metabolite concentrations were measured at the end of each diet phase using gas chromatography/time-of-flight mass spectrometry and ultra-high pressure liquid chromatography/quadrupole time-of-flight tandem mass spectrometry. Participants were provided, in random order, with diets enriched in three carbohydrate types (simple carbohydrate (SC), refined carbohydrate (RC) and unrefined carbohydrate (URC)) for 4.5 weeks per phase and separated by two-week washout periods. Data were analyzed using partial least square-discrimination analysis, receiver operating characteristics (ROC curve) and hierarchical analysis. Among the known metabolites, 3-methylhistidine, phenylethylamine, cysteine, betaine and pipecolic acid were identified as biomarkers in the URC diet compared to the RC diet, and the later three metabolites were differentiated and compared to SC diet. Hierarchical analysis indicated that the plasma metabolites at the end of each diet phase were more strongly clustered by the participant than the carbohydrate type. Hence, although differences in plasma metabolite concentrations were observed after participants consumed diets differing in carbohydrate type, individual variation was a stronger predictor of plasma metabolite concentrations than dietary carbohydrate type. These findings limited the potential of metabolic profiling to address this variable.

## 1. Introduction

Cardiometabolic disorders affect nearly half of U.S. adults [1]. Evidence from both observational and interventional studies supports the premise that diet quality plays an important role in mediating cardiometabolic risk factors (CMRFs) and subsequent cardiovascular disease event rates [1]. As a modifiable behavior, it is important to have the ability to characterize diet quality and make recommendations for the prevention and treatment on the basis of that assessment. For the most part, current diet assessment tools rely on subjective approaches, such as food frequency questionnaires (FFQ) and 24 h food recalls. These assessment tools are vulnerable with respect to inaccurate and biased reporting misestimates due to incomplete food composition tables and lack of replicability [2]. To date, no objective tool is available to evaluate diet quality. 

Metabolomics is a promising approach for objectively assessing diet quality. It relies on quantifying small molecules in a biological system. Several studies support the concept that metabolites reflect responses to different foods and dietary patterns [3,4,5,6]. For example, plasma and urine metabolites, such as alkylresorcinols, and carnosine and 3-methylhistidine have been proposed as biomarkers for whole grain [4,7] and animal protein [8,9], respectively. The SYSDIET Healthy Nordic Diet trial explored the use of untargeted metabolomics to identify plasma profiles associated with a healthy Nordic diet. Compared to the control diet, the healthy Nordic diet resulted in higher concentrations of plasma pipecolic acid, alkylresorcinols (alkylresorcinol isomer C17:0, C19:0 and C21:0), and betainized compounds, which were associated with higher fiber and whole grain intakes compared to the control diet [10,11]. 

Little data were identified that directly compared metabolite profiles in response to diets differing in carbohydrate type: simple (SC), refined (RC) or unrefined carbohydrate (URC). To address this gap, we conducted a secondary analysis using an untargeted metabolomics approach to characterize metabolite profiles in archived fasting plasma samples from a randomized controlled cross-over feeding trial [12] in humans that was designed to assess the impact of different types of carbohydrate on CMRFs. Our hypothesis was that diets enriched in SC, RC or URC would result in unique plasma metabolite profiles. 

## 2. Results

### 2.1. Characteristics and Response of Study Participants

The mean ± S.D. age of the participants was 65 ± 8 y, mean body mass index was in the overweight category (29.8 ± 3.2 kg/m^2^) and six of the participants were females (Table 1). Consistent with the eligibility criteria, participants had elevated LDL-C concentrations (mean 3.5 ± 0.7 mmol/L). Other characteristics were within normal ranges. As previously reported [12], fasting serum total cholesterol, LDL-C and non–high-density lipoprotein (HDL) cholesterol concentrations were higher after the RC diet compared with SC and URC diets (*p* < 0.01). No significant diet effects were observed for the other CMRFs (Appendix A).

### 2.2. Metabolites Profiling

Of the top 20 metabolites (Table 2), the first, second and third components of the PLS-DA explained 13.7%, 10.1% and 5.8% of the variance, respectively (Appendix A). Of those, 36.7% were lipid-related (*n* = 22), 20% were amino acids (*n* = 12), 17% were xenobiotics (*n* = 10), 13.3% were phospholipid-related (*n* = 8) and 13% were carbohydrate, carboxylic acids, ketone body and vitamin related (*n* = 8). A full list of metabolites can be found in Appendix A.

### 2.3. Receiving Operating Characteristics (ROC) Curve for Biomarker Analysis

The area under the curves for the top 20 metabolites ranged from 0.50 to 0.85 (Table 3). Three metabolites (cysteine, betaine and pipecolic acid) were significantly different between SC and URC, five metabolites were significantly different between RC and URC (phenylethylamine, cysteine, betaine, pipecolic acid and 3-methylhistidine) (all *p* < 0.05), and no metabolites were significantly different between SC and RC.

### 2.4. Enrichment and Hierarchical Cluster Analyses

Plasma metabolites identified as significantly different in the diet comparisons were further analyzed using metabolite set enrichment analysis. These data indicate that mitochondria beta-oxidation of short chain saturated fatty acids, beta-oxidation of very long chain unsaturated fatty acids and fatty acid biosynthesis were significantly upregulated after participants consumed SC relative to RC or URC diets (Table 4). 

Hierarchical cluster analysis showed that plasma metabolites were more strongly clustered by participant than type of dietary carbohydrate in the diet (Appendix A). Of the hierarchical clustering by diet, there was considerable heterogeneity among participants. For five participants, the plasma metabolite profiles during the URC diet phase differed from the SC and RC diets. For three participants, the metabolite profiles during the SC diet differed from the RC and URC diets. For two participants, the metabolite profiles during in the RC diet differed the SC and URC diets.

### 2.5. Correlations between Top 5 Metabolites and Cardiometabolic Risk Factors

No significant associations were identified among the top five metabolites in the SC, RC and URC diets and lipoprotein CMRFs (Appendix A). For the SC diet, although there were positive trends for pipecolic acid versus total cholesterol and HDL-C concentrations (pipecolic acid and total cholesterol: r = 0.599; *p* = 0.068; CI: −0.0499–0.8920 and pipecolic acid and HDL-C: r = 0.600; *p* = 0.067; CI: −0.0478–0.8294), these differences had *p*-values above 0.05. A similar observation was made for cysteine and triglyceride concentrations (r = 0.590; *p* = 0.073: CI: −0.063–0.889).

## 3. Discussion

Based on the VIP scores of the known metabolites, amino acid-related metabolites including phenylethylamine, pipecolic acid, cysteine, betaine and 3-methylhistidine were the major metabolites that were higher at the end of the URC diet phase relative to the SC and UC diet phases. The metabolite profiles were relatively similar at the end of the SC and RC diet phases. An enrichment analysis indicated that pathways associated with fatty acid oxidation and biosynthesis were upregulated at the end of the SC compared to RC and URC diet phases. The latter two diet phases were similar. ROC analysis indicated that plasma betaine, phenylethylamine, cysteine, pipecolic acid and 3-methylhistidine increased in response to the URC diet phase compared to the SC and RC diet phases.

Two metabolites, betaine (precursor choline) and pipecolic acid (precursor L-lysine), were identified as potential biomarkers for the diet enriched in unrefined carbohydrates. One prior study that assessed the effects of dietary whole grains on plasma betaine concentrations reported a similar finding [13]. Another prior study that assessed the effects of dietary whole grains on plasma pipecolic acid also reported a similar finding [14]. However, other studies failed to observe this effect [10,11,15,16]. Plasma betaine and pipecolic acid are primarily derived from endogenous gut microbial synthesis. In the current study, the association of dietary whole grains with plasma betaine and pipecolic acid is unlikely due to an effect of the diet interventions on gut microbiota composition. We have previously reported that based on the Shannon index of alpha diversity, there was no significant differences among the SC, RC and URC diets [17]. Using the UniFrac distance to calculate beta-diversity, there was likewise little evidence of structural ecological shifts in response to dietary carbohydrate type. In light of the data reported herein, these conclusions have been corroborated. A prior study using a similar experimental design reported favorable metabolic parameters and increased diet-derived microbial metabolites when participants consumed whole grain compared to refined grain diets, but little impact on microbial composition is observed [18]. Together, these data suggest that the change in microbial metabolites may not be directly associated with microbial population. Thus, additional studies are warranted to address this discrepancy. In the current study, principal coordinate analysis suggested that samples clustered on the basis of participant. Hierarchical cluster analysis confirmed that the plasma metabolite concentrations differed among the diet phases clustered by participant rather than dietary carbohydrate type.

Metabolite set enrichment analysis indicated that pathways associated with fatty acid synthesis and oxidation were more active in participants after they consumed SC diets compared to the RC and URC diets. Pathway analysis also suggested that mitochondria beta-oxidation of short chain saturated fatty acids and very long chain fatty acids were significantly upregulated in response to this active fatty acid biosynthesis. However, plasma fatty acid synthesis and oxidation associated metabolites did not differ among three diets. Of note, we have previously reported similar findings for fecal concentrations of short chain fatty acids [18].

Plasma cysteine, 3-methylhistidine and phenylethylamine were higher after participants consumed the diet enriched in URC compared to SC or RC diets. This is unexplained at this time because these metabolites tend to be associated with dietary animal protein [8,9]. Previous studies reported that plasma L-alanine, carbon-13, D-glucose and 2-piperidinone are potential plasma biomarkers of sugar sweetened beverage intake [19,20]. Similar findings in terms of the SC diet were not observed in the current study. Whether the difference between the two studies was due to measures in fasting versus postprandial state or form of simple carbohydrate remains to be determined.

The aim of this study was to determine the effect of diets differing in type of carbohydrate on plasma metabolites. The approach used was to design intervention diets with a similar amount of total carbohydrate but that differed in type. This was accomplished by incorporating foods made with either simple, refined and unrefined carbohydrate. Because dietary fiber is a constituent of unrefined but not simple or refined carbohydrates, the fiber content of the diets differed. Had we equalized the amount of fiber by adding isolated fiber to the simple and refined carbohydrate diets, a different experimental question would have been addressed. From a translational perspective, it would have had less clinical relevance. A limitation of this approach is we could not isolate the putative factor to attribute the observed effects. Because we previously reported that there was little effect of carbohydrate type on the gut microbiome or fecal short chain fatty acid concentrations, it is unlikely that fiber was a major factor [17]. What was observed is that despite a lack of effect on the gut microbiome, there were differences in several plasma metabolites induced by carbohydrate type that should be further investigated. Another limitation of the study is that the power calculations were based on the primary outcomes of the parent study and not the metabolic component. Data with which to conduct power calculations for future studies are now available.

An unexplained observation was that, in response to the simple carbohydrate diet, the plasma metabolite profile indicated that fatty acid oxidation and fatty acid biosynthesis pathways were activated. Although this appears paradoxical, from a metabolic perspective, it is possible that a rapid influx of glucose from simple carbohydrate stimulated fatty acid biosynthesis, and in response, to maintain lipid homeostasis, homeostatic mechanisms for lipid metabolism responded to the metabolic challenges.

A strength of this study is that the potential introduction of inter-individual variability among participants was minimized due to the nature of a randomized crossover-controlled design. All food and beverages were provided to the participants, minimizing deviations in the feeding protocol during the diet intervention phases. The intervention period, 4.5 weeks, has previously been shown to be adequate for changes in CMRF and microbiota composition induced by dietary perturbations [18]. A combination of three types of platforms covering a wide range of metabolites was used to analyze the plasma samples. Data acquisition, data processing and analysis were well controlled and had low analytical coefficients of variations. A limitation of this study is that only known metabolites were analyzed, and the possibility cannot be ruled out that unidentified metabolites may differentiate SC, RC and URC diets. The dearth of data available from similar diet intervention trials limited our ability to interpret the data relative to prior work. The findings of this study cannot be generalized beyond older adults with elevated LDL-C concentrations. Additional studies with larger sample size are warranted to explore these observations further.

## 4. Materials and Methods

### 4.1. Study Participants and Design

A randomized controlled crossover feeding trial was conducted that included three diet phases (SC, RC and URC), and each was 4.5 week in duration with a minimum 2-week wash-out period between diet phases. The randomization sequence for each participant was generated by the statistician before the start of the study according to a block design, and assignment was based on enrollment date and time. Investigators and laboratory personnel were blinded to the random order. Baseline characteristics were collected during the site visit. Detailed information regarding study population, screening, and participant recruitment were published previously [12]. The primary recruitment criteria were men and post-menopausal women >50 years with a low-density lipoprotein-cholesterol (LDL-C) concentration >3.37 mmol/L. Detailed descriptions of the parent study design and results, including serum lipoproteins, ex vivo fractional cholesterol efflux, adipose tissue gene expression, and ex vivo cytokine secretion, have been previously reported [12]. Of the original 11 participants who consented to optional tissue banking, unthawed plasma samples stored at −80 °C were available for 10 participants. The parent study was conducted between 2012 and 2015 in accordance with the Declaration of Helsinki guidelines, and all procedures were approved by the Institutional Review Board of Tufts University/Tufts Medical Center. The original trial was registered at clinicaltrials.gov as **NCT01610661 (accessed on 11 June 2022)**.

### 4.2. Diet Intervention

The SC diet provided 63.9% total energy as carbohydrate, 13.4% as protein and 22.7% as fat; the RC diet provided 60.8% as carbohydrate, 14.2% as protein and 24.7% as fat; the URC diet provided 61.7% as carbohydrate, 14.7% as protein and 23.6% as fat (Appendix A). Primary sources of carbohydrate for the SC diet were foods containing sucrose and high-fructose corn syrup. Primary sources of carbohydrate for the RC diet were from white rice, bread and pasta. Primary sources of carbohydrate in the URC diet were whole grain versions of those in the RC diet. Participants were instructed to consume all the food provided without additional foods and beverages except for water. The diet composition was determined by chemical analyses (Covance Laboratories Inc., Madison, WI, USA). Study menus have appeared previously [12]. As previously reported, the RC diet resulted in higher fasting serum LDL-C and non-HDL-C concentrations than the SC or URC diets [12].

### 4.3. Untargeted Metabolomics

Untargeted metabolomics analysis was performed on the plasma samples using gas chromatography/time-of-flight mass spectrometry (GCTOF MS) and ultra-high pressure liquid chromatography/quadrupole time-of-flight tandem mass spectrometry (UHPLC-QTOF MS/MS) by the West Coast Metabolomics Center at University of California, Davis. To ensure data quality, automatic liner exchanges were applied after each set of 10 injections to minimize sample carryover for highly lipophilic compounds, and pooled and blank samples were included every 10 and 50 samples.

#### 4.3.1. Primary Metabolites Extraction, Data Acquisition and Processing

The sample preparation and method for GCTOF MS were described previously [8,21,22]. For data processing, raw data were processed using ChromaTOF (LECO Co., St. Joseph, MI, USA) and BinBase databases. Automatic mass spectral deconvolution and peak detection were set at signal/noise levels of 5:1 throughout the chromatogram, and apex masses were reported for use in the BinBase algorithm.

#### 4.3.2. Complex Lipids and Biogenic Amines Extraction, Data Acquisition and Processing

The protocols for lipids and biogenic amines extraction and sample analyses were described previously [8,22]. Lipidomic data were processed using Mass Hunter qualitative analysis, Mass Profiler Professional and Mass Hunter quantitative analysis (Agilent Technology, Santa Clara, CA, USA) for peak alignment and metabolite detection, and Lipidblast was then used for matching MS/MS libraries [23]. Data for biogenic amines were processed by the free mzMine 2.0 software for peaks identification [24], NIST14/Metlin/MassBank online libraries for metabolite identification and Mass Hunter quantitative analysis (Santa Clara, CA, USA) for quantitative results.

### 4.4. Clinical Laboratory Measures

As described previously [12], fasting serum concentrations of total cholesterol (TC), triacylglycerol (TG) and high-density lipoprotein-cholesterol (HDL-C) concentrations were measured using an AU 400e automated analyzer (Beckman Coulter, Brea, CA, USA) with enzymatic reagents (assay coefficient of variation <3%). LDL-C concentrations were calculated using the Friedewald equation [25]. Non-HDL-C was calculated using the following formula: TC-HDL-C. Very low-density lipoprotein-cholesterol (VLDL-C) concentrations were estimated by dividing TG concentrations (mmol/L) by 2.2. Fasting serum glucose concentrations were measured using an enzymatic method (Beckman Coulter).

### 4.5. Statistical Analysis

Study participant characteristics are presented as means and standard deviations (S.D.) for continuous measures and proportions for categorical measures. 

The Appendix A summarized the differences in CMRF between diet phases was adapted from the parent study [12], in which a repeated measured One-way ANOVA was conducted, followed by Tukey-Kramer post hoc analysis, using SAS for Windows (Version 9.4; SAS Institute, Cary, NC, USA) with additional adjustments for diet phase, sequence, age, body mass index and sex). The continuous variables for CMRF in Appendix A were presented as mean (95% confidence interval).

For metabolomics data processing, a total of 3028 metabolites were detected, of which 841 were identified. Pooled samples were used to normalize the data for those metabolites, which were then imputed with the minimum detectable values of less than 80%. Using this criteria, 12 known metabolites were excluded. Data were log-transformed and filtered by the interquartile range for the 829 known metabolites prior to statistical analysis. 

Partial least-square discriminant analysis (PLS-DA) was used to assess the importance of known metabolites using variable importance in projection (VIP) scores. The metabolites were then ranked from the highest to the lowest after using a cutoff 1.7 (Appendix A). The top 20 metabolites, with a VIP > 2.11, were the focus of this report. Receiver operating characteristic (ROC) curves were used to determine the accuracy of the potential biomarkers. Area under the curves and confidence intervals were computed using a cutoff of 0.7 as a threshold to select metabolites. Mapping was performed using metabolite set enrichment analysis, in which three two-group comparisons (SC vs. RC; SC vs. URC; RC vs. URC) were used to compare the metabolic pathways associated with SC, RC and URC diets (0.05/829, *p* < 6.03 × 10^−5^; FDR < 0.05). Hierarchical clustering analysis was used to determine the impact of carbohydrate type on metabolite clusters. Pearson’s correlation and Ward’s linkage were applied for distance measure and linkage method, respectively.

Pearson’s correlation coefficients were calculated to assess relations between diet associated metabolites (*n* = 5) and selected CMRF using R (version 3.6.3; Vienna, Austria) and MetaboAnalyst (version 5.0; https://www.metaboanalyst.ca/home.xhtml; accessed on 18 November 2021) [26]. Statistical significance was defined as two-tailed α ≤ 0.05.

## 5. Conclusions

Although differences in plasma metabolite concentrations were observed after participants consumed diets differing in carbohydrate type, individual variation was a stronger predictor of plasma metabolite concentrations than dietary carbohydrate type. These findings limited the potential of metabolic profiling to address this variable.

## Figures and Tables

**Table 1 metabolites-12-00547-t001:** Baseline characteristics of the study participants.

Variables	Participants (*N* = 10)
Age, y	65 ± 8
Female, n (%)	6 (60%)
Weight, kg	85 ± 12
Body Mass Index, kg/m^2^	29.8 ± 3.2
Fasting glucose, mmol/L	5.6 ± 0.6
Total cholesterol, mmol/L	5.6 ± 0.9
VLDL-C, mmol/L	0.8 ± 0.3
LDL-C, mmol/L	3.5 ± 0.7
HDL-C, mmol/L	1.3 ± 0.3
Triacylglycerol, mmol/L	1.7 ± 0.6

All values are presented as mean ± SD. To convert to mmol/L, for total cholesterol, VLDL-C, LDL-C and HDL-C multiply by 38.67 and, for triacylglycerol, multiply by 88.54. HDL-C, high-density lipoprotein-cholesterol; LDL-C, low-density lipoprotein-cholesterol; VLDL-C, very low-density lipoprotein-cholesterol.

**Table 2 metabolites-12-00547-t002:** Top 20 metabolites with the highest Variable Importance in Projection (VIP) ^1^ score.

Metabolite	Pathway Involved	VIP Score ^1^
Phenylethylamine	Amino acid	3.03
Cysteine	Amino acid	3.00
Betaine	Xenobiotics	2.84
Pipecolic acid	Amino acid	2.83
TMAO	Amino acids	2.57
3-Methylhistidine	Amino acids	2.49
PC 38:3	PC/lipid metabolism	2.47
TG 42:0	lipid metabolism	2.45
TG 51:1(TG 16:0_17:0_18:1)	Lipid metabolism	2.45
Conduritol-beta-epoxide	Xenobiotics	2.37
N-acetylglycine	Amino acids	2.37
TG 45:1(TG 12:0_16:0_17:1)	Lipid metabolism	2.29
PI 36:4	PI/Lipid metabolism	2.25
TG 46:2	Lipid metabolism	2.24
TG 44:0	Lipid metabolism	2.23
Pipecolinic acid	Amino acids	2.23
Coniferyl aldehyde	Xenobiotics	2.17
TG 54:5(TG 18:1_18:2_18:2)	Lipid metabolism	2.15
3-hydroxybutyric acid	Ketone/Lipid metabolism	2.12
LPC 20:3	LPC/Lipid metabolism	2.11

^1^ Variable Importance in Projection (VIP) score was calculated using partial least-squares discrimination analysis. This table shows top 20 plasma metabolites with the highest VIP scores. LPC, lysophosphatidylcholine; PC, phosphatidylcholine; PI, phosphatidylinositol; TG, triacylglycerol; TMAO, trimethylamine N-oxide.

**Table 3 metabolites-12-00547-t003:** Area under the curve–receiver operating characteristics (AUC-ROC) curve for the top 20 plasma metabolites.

Metabolites	SC vs. RC		SC vs. URC		RC vs. URC	
	AUC	FC	AUC	FC	AUC	FC
Phenylethylamine	0.83	0.16	0.70	0.50	0.79	1.06 *
Cysteine	0.56	−0.12	0.84	0.70 *	0.86	0.73 *
Betaine	0.58	0.01	0.85	0.32 *	0.80	0.05 *
Pipecolic acid	0.56	−0.02	0.83	0.97 *	0.74	0.38 *
TMAO	0.69	−0.26	0.63	0.29	0.75	0.40
3-Methylhistidine	0.73	0.80	0.58	0.12	0.78	1.08 *
PC 38:3	0.59	0.06	0.65	−0.22	0.73	−0.15
TG 42:0	0.53	−0.47	0.67	−1.21	0.74	−0.45
TG 51:1(TG 16:0_17:0_18:1)	0.68	0.58	0.59	−0.30	0.71	−0.90
Conduritol-beta-epoxide	0.63	1.33	0.66	−1.24	0.74	−0.99
N-Acetylglycine	0.64	0.10	0.63	0.16	0.76	0.84
TG 45:1(TG 12:0_16:0_17:1)	0.60	−0.24	0.51	−0.56	0.74	−0.30
PI 36:4	0.58	2.42	0.60	−0.47	0.73	−0.99
TG 46:2	0.61	−0.99	0.68	−0.49	0.67	−1.79
TG 44:0	0.53	−0.47	0.66	−1.41	0.70	−1.28
Pipecolinic acid	0.50	−0.23	0.74	0.98	0.74	1.99
Coniferyl aldehyde	0.70	−0.28	0.52	−0.24	0.77	0.29
TG 54:5(TG 18:1_18:2_18:2)	0.64	−0.13	0.59	0.14	0.75	−0.22
3-hydroxybutyric acid	0.67	−0.07	0.66	0.76	0.66	−0.29
LPC 20:3	0.61	−0.07	0.63	0.09	0.71	−0.12

AUC-ROC curves were generated (cutoff: 0.7). *, *p* < 0.05. The data shown in the fold change (FC) column are relative to SC or RC. AUC, area under the curve; LPC, lysophosphatidylcholine; PC, phosphatidylcholine; PI, phosphatidylinositol; RC, refined carbohydrate diet; SC, simple carbohydrate diet; TG, triacylglycerol; TMAO, trimethylamine N-oxide; URC, unrefined carbohydrate diet.

**Table 4 metabolites-12-00547-t004:** Active metabolic pathways in participants who consumed the simple carb diet compared to refined and unrefined carb diets.

Pathway	SC vs. RC	SC vs. URC
	FDR	FDR
Mitochondria beta-oxidation of short chain saturated fatty acids	2.69 × 10^−3^	8.27 × 10^−7^
Beta-oxidation of very long chain fatty acids	4.36 × 10^−3^	1.41 × 10^−5^
Fatty acid biosynthesis	7.11 × 10^−3^	1.73 × 10^−3^

The Benjamini and Hochberg procedure was conducted to account for multiple comparisons, and statistical significance was defined as FDR < 0.05. FDR, false discovery rate. No statistically significant pathways were identified for the RC or URC diets.

## Data Availability

The data that support the findings of this study are available from the corresponding authors upon reasonable request.

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
