# Peer review of "Plasma Metabolite Response to Simple, Refined and Unrefined Carbohydrate-Enriched Diets in Older Adults—Randomized Controlled Crossover Trial"

_metabolites, 2022, doi:10.3390/metabo12060547_

Round 1

Reviewer 1 Report

The article titled "Plasma metabolite response to simple, refined and unrefined carbohydrate-enriched diets in older adults – randomized controlled crossover trial" addresses how different dietary carbohydrate sources affect metabolism. One of the strengths of the study is the well-controlled design of cross-over treatments and the broad metabolite coverage using a combination of GC-MS and LC-MS. The remainder of the methodology, including the statistical analysis, is appropriate too. The paper is well-structured and well-written, the presentation of the results is clear, and the conclusions are well-supported by the data. This reviewer requests only one minor correction: the carbohydrate sources need to be indicated for the RC group - currently the SC is listed twice (L82-83). 

Author Response

Dear reviewer,

Thank you very much for your time and consideration. Please see the attachment for the response, and we have addressed the comments point by point.

Thank you again for your time.

Sincerely,

Neil K. Huang

Reviewer 2 Report

The results in this paper give a better understanding of refined and unrefined carbohydrate-enriched diets. I want to accept it after minor revision. However, a more detailed and large-scale investigation should be conducted.

Introduction:

The results of SC, RC, and URC in the previous report [12] and the hypothesis should be described in more detail.

2.2. Diet Intervention:

It is wrong notation “…SC diet were white rice, bread and pasta...”

In the last sentence, it was unclear what the author wanted to mean.

3.4. Hierarchical cluster analysis

Is the word “differed significantly” this has statistical meaning?

Participants 5 and 7 differed significantly from the other participant. Are there any possible factors?

3.5. Correlation between top 5metabolites and CMRFs

In several metabolites, if r>0.576 it seemed significant associations (n=10, p<0.05), and should be discussed.

Author Response

Dear Reviewer,

Thanks again for your time and consideration. Please see the attachment for the responses as we revised the manuscript and responded to the comments point by point.

Thank you!

Sincerely,

Neil K. Huang

Reviewer 3 Report

This is a manuscript that studies a relevant issue, the influence of the type of carbohydrate on the metabolism and the cardiovascular risk.
The authors use a previous study and analyze a reduced number of samples (10) to investigate the serum metabolites in response to three different diets. The metabolomic determinations are well described and the statistical analysis is correct. However, the article has several problems:
1.- Diets differ not only in the type of carbohydrate but also in total sugar content and also in fiber content (the differences in insoluble fiber are huge in the URC group). This clearly could alter the results since the effects are not only due to the different digestibility of the carbohydrates but also the fiber (for example in the production of SCFA) that has a clear effect on the metabolic effects of the diet.
2.- The conclusion of the manuscript is negative since the authors indicate "individual variation was a stronger predictor of plasma metabolite concentrations than dietary carbohydrate type" which is disappointing.
3.- From the results, the authors indicate that in the SC diet the fatty acid oxidation pathways and fatty acid biosynthesis pathways are activated at the same time. This is difficult to understand from a metabolic point of view and needs an extensive discussion to be clearly explained.
4.- In the experimental section (dietary intervention), the experimental groups are named and described differently. For example, a USC group appears.

Author Response

(The authors gave the same response as above.)

Reviewer 4 Report

The manuscript is well written and will provide important knowledge in the field. I do have few minor comments and some suggestions.

Page 2, section 2.2: The primary sources of carbohydrate for RC are missing; or possibly due to some typographical error, SC is mentioned twice for the carbohydrate source. Please double check the statements.

Table 1, caption; Do the authors mean the conversion factor from mmol/L to mg/dl? This should be described.

Discussion: The data shows that TMAO is one of the top plasma metabolites. Even though there is no significant difference among the groups, some discussion about TMAO related to the carbohydrate enriched diets would be useful.

Discussion: There is significant difference in fiber content among the experimental diets, which might impact the findings, thus this should be discussed.

Author Response

(The authors gave the same response as above.)

Reviewer 5 Report

The manuscript by Huang and colleagues shows the results of a cross-over design RCT on a sample of 10 individuals (selected from a previous cardiovascular outcomes study). The intervention consisted of 3 phases of 4.5 weeks each, with a nutritional intervention focused on the modulation of carbohydrate types (simple, refined or unrefined) with two weeks of washout between phases, to evaluate the correlation between type of ingested carbohydrates and metabolites produced. Despite differences in the comparison between the various phases, the decisive determinant seems to be individual and not significantly dependent on the type of nutritional intervention.

The manuscript is smooth and well organized. The description of the experimental procedure is detailed and the topic is of great interest. The uncertainty associated with canonical surveys on eating habits is an important aspect that limits the accuracy of the results in clinical trials and observational studies. The effort made by the authors to propose an investigation that can overcome these difficulties, showing the potential of metabolomics, is evident. However, I have some concerns about the relevance of the study in advancing the state of the art on the subject and unfortunately, some aspects cannot be revised

- A detailed description of the population characteristics at baseline may be helpful. How were the groups organized in the three arms of intervention in the initial phase? How was the crossover between the 3 interventions organised? I realize that the original trial has already been published but I believe that each manuscript must contain all the information useful for evaluating the solidity of the results. The possible baseline differences represent a minor problem in the case of clinical trials with a cross-over design

- It is essential to specify that the results obtained are not transferable to different population samples. The authors selected a cohort of old individuals with metabolic risk factors, so this may differ from young, healthy individuals.

- The individual characteristics of the participants could be used for the evaluation of adjusted correlations between the type of diet and the production of metabolites with a known role, although I am not sure that the low number of participants allows this.

- There is no verification of adherence to the dietary intervention, although this aspect could be less decisive, taking into account that the dishes were standardized and provided by the experimenters

- The low sample size is the aspect that worries me the most. A cross-over clinical trial with only 10 individuals may not have sufficient statistical power to confirm the authors' hypothesis.

- Significant differences between the components of the three different dietary approaches should be provided (p-value in Table S1). However, from Table S1 it emerges that the simple carb diet contains more sugars even starch. At the same time, simple sugars are present in the other two dietary approaches. This mitigates the differences between the various diet phases. It would have been more correct if there were almost exclusively sugars in the simple carb diet and the other two, only complex carbohydrates.

Author Response

(The authors gave the same response as above.)

Round 2

Reviewer 3 Report

Albeit some important points have not been addressed, for example, the relevance of the fiber in the experimental design, the manuscript has been improved. Some explanations regarding the synthesis and degradation of lipids should be elaborated to provide a more plausible explanation.

Reviewer 5 Report

I really appreciate the authors' effort to implement their manuscript. Even if the work done is evident, the important knots of my perplexities remain. As the authors themselves stated, "the sample size was not powered for the metabolomic work". Furthermore, despite the understandable practical limitations for organizing a dietary intervention that can isolate sugars from refined and whole complex carbohydrates, this difficulty compromises the initial assumption proposed by the authors. Furthermore, the sentence in lines 344-346 of the revised manuscript, used several times in response to my requests, is too general to focus the reader on the limitations of the work, first of all, those listed above.